# A Multimodal Research Approach to Assessing the Karst Structural Conditions of the Ceiling of a Cave with Palaeolithic Cave Art Paintings: Polychrome Hall at Altamira Cave (Spain)

**DOI:** 10.3390/s23229153

**Published:** 2023-11-13

**Authors:** Vicente Bayarri, Alfredo Prada, Francisco García

**Affiliations:** 1GIM Geomatics, S.L. C/Conde Torreanaz 8, 39300 Torrelavega, Spain; vicente.bayarri@gim-geomatics.com; 2Polytechnic School, Universidad Europea del Atlántico, Parque Científico y Tecnológico de Cantabria, C/Isabel Torres 21, 39011 Santander, Spain; 3Museo Nacional y Centro de Investigación de Altamira, Marcelino Sanz de Sautuola, S/N, 39330 Santillana del Mar, Spain; alfredo.prada@cultura.gob.es; 4Department of Cartographic Engineering, Geodesy and Photogrammetry, Universitat Politècnica de València, Camino de Vera, s/n, 46022 Valencia, Spain

**Keywords:** ground penetrating radar, data integration, mapping, karst system, rock discontinuities, cultural heritage, rock art, preventive conservation, geomatics, cultural management

## Abstract

Integrating geomatics remote sensing technologies, including 3D terrestrial laser scanning, unmanned aerial vehicles, and ground penetrating radar enables the generation of comprehensive 2D, 2.5D, and 3D documentation for caves and their surroundings. This study focuses on the Altamira Cave’s karst system in Spain, resulting in a thorough 3D mapping encompassing both cave interior and exterior topography along with significant discontinuities and karst features in the vicinity. Crucially, GPR mapping confirms that primary vertical discontinuities extend from the near-surface (Upper Layer) to the base of the Polychrome layer housing prehistoric paintings. This discovery signifies direct interconnections helping with fluid exchange between the cave’s interior and exterior, a groundbreaking revelation. Such fluid movement has profound implications for site conservation. The utilization of various GPR antennas corroborates the initial hypothesis regarding fluid exchanges and provides concrete proof of their occurrence. This study underscores the indispensability of integrated 3D mapping and GPR techniques for monitoring fluid dynamics within the cave. These tools are vital for safeguarding Altamira, a site of exceptional significance due to its invaluable prehistoric cave paintings.

## 1. Introduction

Prehistoric art, an extraordinary manifestation documenting detailed facts and coexisting animal species in Palaeolithic times, is often found in caves within evolving karst systems housing unique microhabitats. These special and fragile artworks have faced preservation challenges due to tourism pressures. The Upper Palaeolithic, spanning 40,000–10,000 years ago, marked a significant era characterized by the emergence of anatomically modern humans (*Homo sapiens*) and extraordinary advancements in technology, art, and culture. This period witnessed the flourishing of cave paintings, sculptures, and personal ornaments, reflecting an evolving capacity for abstract thought and symbolic representation.

The Altamira cave has one of the most important pictorial and artistic cycles of the Upper Palaeolithic period. The cave was used for several periods during 22,000 years of occupation, from about 36,500 years ago until 13,000 years ago, when the main entrance to the cave was sealed by a collapse [1,2]. This cave is in Santillana del Mar, Cantabria, Spain (Figure 1), and was declared a World Heritage Site by UNESCO in 1985. The recognition of such universal values brought with it the responsibility to ensure its preservation [3,4,5,6,7].

The cave is part of a common geological and landscape environment. Altamira, with other caves close to it, such as the cave of Stalactites and the cave of Castañera, arise as consecutive episodes in the evolution of the karst, with Altamira, in the uppermost part of the karst, being the oldest. Understanding its relationship with the environment is fundamental to explaining the influence this system of karst galleries can have on the physical and chemical equilibria of Altamira and the stability of the microclimatic conditions in the cave [8].

The cave embodies a unique karstic environment essential to understanding its preservation. It is within a senile Pliocene-age karst that features a tubular, polygenic structure affected by geological and structural processes, notably fracturing, resulting in cavities that are formed through gravitational collapses [9]. Due to its proximity to the surface, the cave remains reliant on rainfall infiltration for water supply, with minimal in-cave circulation due to the rock’s low permeability. The network of fissures, fractures, and detachments present from the top of the cave to the ceilings themselves affect the general conservation of the cave as they present a risk to its stability due to the possibility of gravitational collapse [10].

The Polychrome Hall’s ceiling within Altamira cave adds another layer of complexity. It is formed by a limestone stratum sloping to the south, separated from the surface by a modest limestone layer. The surface hosts prehistoric paintings but is affected by microcorrosion processes, forming small millimetre-sized domes, while clay films with sporadic carbonate content adhere to the roof. Most intriguingly, the cave is above the maximum reach of lateral infiltrated water from higher areas, emphasizing that the water source for this unique environment is primarily confined to localized rainfall infiltration. Altogether, Altamira’s intricate geological and environmental features underscore the paramount importance of studying its immediate surroundings to preserve its cultural and natural heritage.

The preservation of cave paintings at Altamira under optimal conditions due to low natural infiltration and entrance collapse was disrupted by human discovery and visits, irreversibly changing the cave’s ecosystem [11,12]. Natural alteration in the paintings began at creation and has remained consistent, with current conservation concerns tied to inevitable geological and hydrogeochemical deterioration processes [13]. Altamira’s Polychrome Hall paintings on the ceiling face added complexity in preservation and the application of documentation techniques like GPR, photogrammetry, and hyperspectral remote sensing owing to their horizontal arrangement and vulnerability to washing. The main deterioration affecting the conservation of the paintings is the loss of pigment through washing, which has occurred over millennia and has led to the alteration or disappearance of parts of the paintings, especially in the southern section of the ceiling in the Polychrome Hall [14,15].

These pigment loss processes are associated with several factors: infiltration, condensation and composition water, relief of the support, and relationship with the fracture network, which allow natural ventilation of the Polychrome Hall and the cave [16,17]. Nowadays, these loss processes are monitored using geomatics techniques [18,19,20,21] such as photogrammetry and hyperspectral remote sensing, which help not only to document pigment losses but also to try to recover veiled areas or to create a chronological stratigraphy of the art [22].

The existence of an important network of fractures and delamination present from the top of the cave to the cave ceilings themselves affected and continues to affect its stability. The gravitational landslides documented in recent times highlighted the need for building artificial walls in areas that showed significant instability in the 1940s and 1950s. Among these areas is the Polychrome Hall itself, where the ceiling is crossed by a very relevant central crack. While the aforementioned retaining walls were being built, this large central fracture, as well as other adjacent ones, were sealed with hydraulic mortar, reactivating new and relevant deterioration processes.

Water supply to the Polychrome Hall comes from direct infiltration of rainwater through the fractures; lateral contributions of infiltrated water are small.

The overlying thickness of the Polychrome Hall bedrock is between 7 and 9 m (Figure 2), with the thickness of the soil cover varying between 5 and 25 cm.

Accurate georeferencing of the interior of the cave with the exterior is crucial to preserving the geological formations, habitats, artifacts, and rock art in the cave. Knowing this information allows for the establishment of protection zones, tracking the internal microclimate of the cave, mapping safe zones, and helping with research, hence helping to achieve effective and sustainable conservation [23].

Nowadays, the use of geophysical surveys is a possible alternative to direct research methods for studying the internal rock structure in karstic complexes [24,25,26,27], particularly for studies of karst formations that feature prehistoric art [28,29,30]. Geophysical surveys are non-invasive and can accurately map discontinuities in the shallow subsurface [31,32,33].

Ground-penetrating radar (GPR) reflection profiles are commonly used to examine the internal structure of rock masses, encompassing rock matrix blocks and discontinuities. These media are typically characterized as mechanically discontinuous, anisotropic, and heterogeneous. In recent decades, GPR reflection profiles have also proven valuable for correlating contacts between stratigraphic units or identifying structural breaks (discontinuities) within geological formations, thus enhancing our understanding of subsurface complexities [34,35]. GPR is a non-destructive method that provides continuous, high-resolution information about the internal rock structure and its physical properties, both vertically and laterally [36,37,38,39,40,41].

The main objective of this study was to systematically map the discontinuities within the strata overlying the Polychrome Hall to evidence their role as conduits for flow processes connecting the cavity with the external environment, an aspect previously considered only as a hypothesis. This article showcases the multidisciplinary tasks carried out to model both the thickness of the overlying layer and the water supply basin in the Altamira cave’s Polychrome Hall. The aim was to confirm the consistency of the stratigraphic sequence, precisely characterizing and mapping the fractures within each package and pinpointing significant vertical fissures that facilitate direct water infiltration, thereby endangering the preservation of the invaluable paintings.

In the following parts of the study, readers can expect to delve into a comprehensive exploration of the Polychrome Hall within Altamira cave. Section 2 offers detailed insight into the technical aspects of Ground-Penetrating Radar (GPR) methodology and the specific parameters of 400 MHz and 900 MHz antennas used in the study; these serve as crucial tools for understanding the geological and hydrological dynamics in the cave. Section 3 takes us through the processing flow of GPR data and the application of hydrological tools to map water infiltration and the interconnections between the cave’s exterior and interior environments. Additionally, it delves into the internal stratigraphy near the cave’s surface, providing valuable information for conservation and understanding of the rock art’s context. Finally, in Section 4, readers are presented with a detailed discussion of the findings. The study emphasizes the importance of elevation models, discusses the complex spatial arrangement of discontinuities in the cave, and highlights the need for ongoing conservation efforts to protect the cave’s valuable art and mitigate risks of damage. Overall, these three parts offer a comprehensive journey into the geophysical and geological aspects of Altamira’s Polychrome Hall, shedding light on its historical significance and the measures necessary for its preservation.

## 2. Materials and Methods

Caves are formed by the dissolution of limestone rock due to slightly acidic water and involve chemical and geological processes as well as tectonic forces and atmospheric influences. These caves are irregular in shape and size, requiring the use of various georeferenced remote sensing techniques to map them and obtain more information about this valuable heritage [42].

### 2.1. General Workflow

The data integrated into the model requires, first, a reference frame, which was generated using a Global Navigation Satellite System (GNSS) [43]. GNSS is a technology that enables global geospatial positioning through a network of satellites [44]. In 2013, a geodetic reference system was established using TOPCON Hyper II receivers [45] and the European Terrestrial Reference System 1989 (ETRS89). Topcon Tools software [46] was used to calculate observed phase changes, resulting in a reference frame with a mean accuracy of 1.7 cm in determining the coordinates of the vertices located outside the cave.

Having an accurate cartographic model of a cave is important when predicting knowledge-based models relying on georeferenced parameters like microbiological, climatic, hydrochemical, and geomorphological data [47]. These parameters often show significant correlations with each other and with external environmental factors. A closed back-and-forth traverse [48] spanning a total length of 430 m and involving 16 traverse stations was conducted within the cave. The angular closure error was 0.0218 degrees and the linear closure error was small (X = −0.001 m, Y = −0.005 m, Z = 0). To establish the georeferenced system for 3D laser scans, 66 checkerboards positioned throughout the cave were radiated from traverse vertices. This reference system was pivotal for the precise georeferencing of the 300 scans conducted using a FARO FOCUS X-130 [49] during the field campaign in December 2013. The scans were calibrated using spheres as tie points and checkerboards as references, resulting in an accuracy of 2.7 mm for 95% of the points to which the interior photogrammetry [47] was supported.

The exterior model of the cave was generated by an Unmanned Aerial Vehicle (UAV). UAVs have become integral tools in surveying and mapping [50,51,52], offering efficient and reliable means for assessing large areas in significantly less time compared with traditional techniques such as Topographic Total Stations (TTS) and GNSS. Moreover, UAVs are nowadays also integrated with thermal cameras and lamps, enabling them to perform thermographic inspections, hence expanding their capabilities in various applications [53]. In our study, we harnessed the capabilities of a TOPCON Intel Falcon 8+ Drone [54] equipped with a Sony A7 R Mark II full-frame camera [55] incorporating a Sony Sonnar T* FE 35 mm F/2.8 lens. By implementing a flight planning system, we achieved a ground sample distance (GSD) of 2 cm across the extensive cave environment (40 hectares). Over 4800 images were collected, with 52 Ground Control Points (GCPs) measured via RTK-GNSS to establish flight references. An additional 40 points were measured for quality control, yielding a mean control point error of 1.76 cm.

These photogrammetric flights enabled the acquisition of diverse spatial data, including 2D orthoimages, 2.5D Digital Surface Models (DSMs) transformed into Digital Elevation Models (DEMs) through tree removal, and 3D datasets encompassing point clouds and comprehensive models of both the cave environment (40 hectares) and its adjacent areas (4 hectares). The 3D model of the cave was composed of 12 million triangles textured with eight 8192 × 8192-pixel textures. Simultaneously, the model of the neighbouring area featured four million polygons and four 8192 × 8192-pixel textures serving as a foundation for data integration. Both models were exported in .Obj file format for seamless incorporation with more data sources, underscoring the versatility and precision of drone-assisted surveying in our study. All the details on integrating 3D TLS, UAVs, and exterior GPRs (Figure 3) can be found in [56].

Previously, in 2014, a photogrammetry campaign was carried out to document the ceiling of the Polychrome Hall [57], independently of the laser scanning process, to achieve sub-millimetre resolution documentation. Besides the geometrical quality of the image, a camera capable of capturing true R-G-B colour was chosen, as the coded images were also to be used for colour analysis. For this purpose, a Hasselblad H4 D-200 MS, which can capture 8176 × 6132-pixel frames with the HC 3.5/35 and HC 4/28 lenses, was used.

The minimum resolution was set to 16 pixels/mm^2^ and the illumination system consisted of two 4800 Kelvin degree LED flat screens with an illuminance of 980 lux each (composed of 1024 LED units), which were covered with a Rosco Cinegel 3000 Tough Rolux (Roscolab Limited, London, UK) plastic diffuser.

Photogrammetric support was carried out by detecting homologous points extracted from the point clouds captured by the 3D laser scanner. A total of 80 homogeneously distributed points were used throughout the hall, half of which were used as control points for model validation. The images were adjusted in blocks and a high-density point cloud of about 11 billion points was generated from them. This was filtered and generalized to another cloud of about 3.5 billion points from which a high-resolution 3D digital model of the ceiling of about 200 million polygons was used to generate a 6 Gigapixel orthoimage and a reduced 4 million polygon version used for the integration.

### 2.2. Ground-Penetrating Radar

In recent decades, GPR reflection profiles have been correlated with contacts between stratigraphic units or with structural breaks (discontinuities) in geological media [42,43]. GPR studies of the internal structure of rock masses focus primarily on their discontinuities, including fractures, joints, veins, detachments, and stratification planes [58,59,60,61]. The karst system of the Altamira cave is in a senile state and is characterized by multiple discontinuities in its different strata (Figure 4) [11,62]. The information is collected by transmitting electromagnetic energy into the subsurface and recording the reflected, diffracted, or critically refracted energy from the electromagnetic discontinuities. Discontinuities in a rock mass represent electromagnetically active contacts/interfaces to the GPR, where part of the radar pulse energy is reflected to the surface, producing a radar record as a profile (radargram). A radargram can be represented in grayscale or colour, where different shades or colours correspond to different magnitudes of the recorded wave amplitudes [63,64].

The penetration depth at which the GPR technique can obtain records/images of discontinuities within a rock is strongly influenced by the electromagnetic parameters within the rock mass (dielectric permittivity, electrical conductivity, and magnetic permeability). These electromagnetic parameters can indicate alterations in the rock and the presence of conducting minerals, porosity, joints, fractures, water content, and fillings within the discontinuities. Penetration is greater in rocks characterized by low electrical conductivities and magnetic permeabilities. Most geological materials and rocks are non-magnetic, thus the dielectric permittivity (dielectric constant) and electrical conductivity, in particular, play a relevant role in amplitude attenuation and effectiveness of the GPR. The attenuation factor increases with increasing conductivity of the medium. Additionally, the electromagnetic wave undergoes geometric attenuation with distance. Reflective/diffractive surfaces also contribute to signal attenuation. In summary, the degree of water saturation, water mineralization, distance, variation in porosity, and number of discontinuities all affect the propagation of electromagnetic waves in a rock mass [63,64,65,66].

The signal penetration and detection resolution of the GPR depends on the frequency of the antenna used. Higher frequency antennas offer better detection resolution, but signal attenuation also increases, resulting in less penetration into the rocks. The choice of antenna type, considering its central emission frequency, is made taking into account the detection–penetration trade-off [64,65].

In the karst of the Altamira cave, an earlier study of the layers overlying the Polychrome Hall observed attenuation of the electromagnetic wave in radargrams obtained using an antenna of 100 MHz from about 5–6 m deep [47], that is, the penetration depth reached with the 100 MHz antenna was not enough to study its full thickness (>9) (Figure 2). This attenuation of the electromagnetic wave in the layers overlying the Polychrome Hall was the main reason GPR profiles were projected on the Polychrome Hall ceiling using 400 MHz and 900 MHz antennas. Thus, radar records that complement the final meters up to the basal surface of the Polychrome Hall ceiling, which could not be studied from its outer surface using the 100 MHz antenna, would be obtained. In addition, the discontinuities and geological layers close to the basal surface of the Polychrome Hall could be studied in detail. Their study is crucial as they support the Palaeolithic paintings.

3D integration of the three UAV, 3DTLS, and exterior GPR models of the cave and its overlying layer was achieved [47]. Nevertheless, constraints associated with time limitations imposed by conservation efforts in the interior of Polychrome Hall prevented the acquisition of a continuous spaced grid that, when interpolated, would have provided the accuracy required for our purposes. As a result, we had to settle for the 2D profiles that we were able to record.

#### 2.2.1. Field Data Acquisition: Polychrome Hall ceiling

A GSSI SIR3000 system was used in this geophysical study. Due to the morphology and dimensions of the Polychrome Hall ceiling, and considering the mobility factors inside it, profiles close to the ceiling were measured using the 400 MHz and 900 MHz central frequency antennas. Data acquisition using the 400 MHz antenna was performed in continuous mode, with a total time window (range) of 100 ns and defined by 512 samples per trace, and with IIR filter 800 MHz (low pass)–100 MHz (high pass). Data acquisition was also performed in continuous mode using the 900 MHz antenna, with a total time window (range) of 35 ns and defined by 512 samples per trace, with IIR filter 2500 MHz (low pass)–225 MHz (high pass). The profiles were transversal and parallel to the central fracture of the Polychrome Hall ceiling. Access to the Altamira cave is restricted to protect its integrity and historical value. Conservation measures have been put in place to limit the time visitors and researchers can stay in the cave to minimize their impact and prevent damage to the paintings. Due to these limitations, 46 GPR profiles 3 m in length were recorded inside the cave and used for the present study.

Recording these GPR profiles allowed us to complement the mapping and detection of discontinuities obtained from the outer surface of the Polychrome Hall using the 100 MHz antenna as described in a previous study [47], as well as record discontinuities with higher resolution in the geological levels closest to its basal surface. Measuring these discontinuities in greater detail is of great interest since the Palaeolithic paintings are located on the basal surface of the Polychrome Hall ceiling.

Several authors have conducted software simulation studies and real measurements of the optimal height of antennas relative to a surface during data collection over the last decades [66,67,68,69,70,71,72,73]. These authors concluded that the best GPR data are obtained when antennas are in contact with the ground or close to the ground. This allows the antennas to couple to the ground and ensures that the maximum amount of energy is transmitted to the subsurface. The higher an antenna is above the ground surface, the more GPR energy is reflected at the interface with the ground, and the less GPR energy is transmitted to the subsurface, greatly reducing the penetration depth. This not only reduces the response amplitude of smaller targets but also decreases the spatial resolution. Furthermore, the height of the antenna depends on its centre frequency. Higher centre frequency antennas need to be closer to the surface, while lower centre frequency antennas can operate with wider gaps between the antenna and the surface. The authors also concluded that the results generally showed optimal performance up to an antenna height of 100 cm.

GPR data were collected with the antennas air-coupled to the ceiling of the Polychrome Hall. A 3-m-long slider, on which the 400 and 900 MHz antennas focused on the ceiling were placed, was chosen due to operative limitations inside the cave (Figure 5a). An odometer was used to accurately reference the GPR profile to the source. A key aspect of the model was the georeferencing of the profiles registered inside the Polychrome Hall, which was achieved by 3D scanning of the sliders where the antennas were placed. In addition, a minimum distance of about 8 cm from the ceiling was ensured to avoid any antenna–ceiling contact that would cause damage to the cave paintings. We had to assume that the height of the antenna would affect the ceiling surface during data acquisition. To minimize this effect and take into account a minimum permitted distance to the ceiling paintings, data acquisition was performed with heights based on the centre frequency of the antenna: 10 cm height for the 1.6 GHz antenna and 25 cm height for the 400 MHz antenna. These heights allowed the antennas to be air-coupled to the ceiling to ensure the maximum possible amount of energy transmitted to the rock mass and to reduce air-coupling effects to obtain accurate and quality GPR records [68].

#### 2.2.2. Data Processing

##### Velocity Estimation

Several hyperbolic reflections are present in different GPR profile sets measured in the ceiling area. The hyperbola fitting method was used to determine the average velocity of the GPR wave, obtaining a value of 0.109 m/ns. Dielectric permittivity (*ε*) was calculated as 7.5 using the simplified equation for low-loss media [64]:(1)$$v=\frac{c}{\sqrt{\varepsilon_{r}}}$$
where *v* is the electromagnetic wave velocity and *c* is the velocity of light in free space (*c* ≈ 0.299 m/ns).

Velocity variations are present in rock masses with discontinuities. Although measuring a single velocity value for data migration is not an optimal procedure, this value was considered acceptable after evaluating the disposition of some diffractions and observing coherence between the thicknesses of geological levels mapped at different depths in the radargrams obtained and those in earlier studies, both on the surface of the thickness of the overlying layer and on the Polychrome Hall ceiling [8,10,11]. This velocity value was used to calculate thicknesses and depths in the carbonate layers based on the parameters of each of the 400 MHz and 900 MHz antennas (Table 1). This average velocity was also used for processing the 2D GPR data in the Kirchhoff migration.

##### Processing Flow

Several post-acquisition radar signal process techniques can be used for GPR datasets based on proposed goals. The RADAN 7.6 software (Geophysical Survey Systems, Inc., GSSI, Nashua, NH, USA) was used in this study. The main phases for processing the GPR raw data (Figure 6) were:

The first phase is the 1D treatment consisting of the following points:As in our case, when the RADAN processing software is used, the initial component of the DC signal—also known as DC shift—as well as the low-frequency trend, is automatically removed. The DC shift is usually the first processing step/filter applied to the data after acquisition [66].Zero-time correction (zero-time adjustment) is applied to remove the delay time in the first reflection to correct for the effect of the distance between the transmitting (T) and the receiving antennas (R) and possible variations in the arrival time of the direct waves.In the second phase, 2D processing was applied to the profiles based on the following points:Background filtering was applied to remove the average values on all traces on the horizontal axis (horizontal bands) based on the antennas used. The average of all traces was calculated by averaging all traces in the profile and subtracting them from each trace starting at time zero.Various bandpass filters were applied depending on the centre frequencies of the antennas used to remove high and low-frequency noise in the vertical and horizontal directions.Linear amplitude gains were applied to compensate for amplitude reduction due to the filters run. This allowed for the highlighting or enhancement of certain amplitudes.Kirchhoff migration was applied to the profiles using the calculated average velocity (0.109 m/ns) to move the reflectors to their true subsurface positions.Time-to-depth conversion was performed for the rock mass based on a calculated average velocity of 0.109 m/ns.

## 3. Results

### 3.1. Polychrome Overlying Layer Hydrology

A series of hydrology tools have been used to model the flow of surface water from the DEM generated with the UAV. The aim is to understand how surface water flows in the area near the Polychrome Hall and to identify possible sinkholes through which water enters the system as well as the layout of the different streams and the extent and shape of the catchment basins.

The Polychrome Hall ceiling orthoimage [56] provides information on the parts of the ceiling that were washed by water in the past and sealed by calcium carbonate, and the areas that are still actively leaking (Figure 5b).

Hydrological tools are used to simulate the movement of water over surfaces. In the area overlying the Polychrome, water flows have been modelled to understand the origins, pathways, and destinations of water (Figure 7a,b). For this purpose, layers were generated using QGIS software [74] for these parameters [75,76]:Drainage Basins: Drainage basins, also known as watersheds, are areas of land from which all surface water flows into a common outlet such as a river. These basins are delineated by the topography of the land, with higher elevations forming the basin boundaries.Catchment Basins: Catchment basins follow the same concept as drainage basins. They are the geographical areas that contribute runoff water to a single point, usually a river or stream.Flow Direction: Flow direction in hydrology and GIS refers to the direction in which water flows across the terrain. It is a crucial part of modelling the movement of water in a digital environment.Watercourses: Watercourses are natural or man-made channels through which water flows.Flow Accumulation Zones: Flow accumulation zones are areas within a watershed where water from surrounding areas converge. These areas are typically identified by analysing the terrain and the direction of the flow of water.Flow Distance: Flow distance is the distance water travels along the terrain to reach a specific point downstream.Sinks: In hydrology, sinks are depressions or low points in the terrain where water can accumulate. These are especially interesting for our case study because they represent the points where all the captured water is introduced into the limestone package.Strahler Method for Watercourse Classification [77,78]: This is a technique used to classify the hierarchical order of watercourses within a watershed. In this method, the smallest headwater streams are designated as first-order streams. When two first-order streams converge, they form a second-order stream, and so on. This classification helps to explain the branching structure of a watershed’s drainage network.

### 3.2. Internal Stratigraphy

The reflectors recorded in the GPR profiles of the Polychrome Hall were correlated with the arrangement of the strata as described in the CSIC report [8] on the detailed lithostratigraphic column representative of the layers in which the Polychrome Hall is found based on the type of lithostratigraphic series of the Santillana area [10,79]. The internal stratigraphy (Figure 8), from the ceiling of the Polychrome Hall to the surface, can be summarised as follows [8]:Polychrome layer: this limestone layer is about 60–70 cm thick and tapers towards the south. The base exhibits hydroplastic deformation, which is significant as the Palaeolithic paintings are at its base, taking advantage of the hydroplastic deformations’ shapes.Dolomitic layer: this dolostone layer (10–25 cm thick) displays varying degrees of dolomitization. A thin loamy-clay intercalation separates it from the Polychrome level.Fissured layer: a limestone layer, about 1.1 m thick, that is visible throughout the cave. It is recognized by its unique alteration pattern characterized by vertical dissolution morphologies associated with joints.Orange layer: this layer consists of thin-bedded limestone (10–20 cm each) with a total thickness between 0.7 and 1 m.Upper layer: this layer alternates between calcarenite and limestone, with thin marl interbeds. It has a total thickness of about 2 m.

### 3.3. Layer Overlying the Polychrome Hall

The accurate georeferencing of the interior 400 MHz antenna and the exterior 100 MHz antenna made the mapping of the layout and shape of the main vertical discontinuities possible (Figure 9). This mapping is the first evidence confirming that the main vertical discontinuities run from near the surface (Upper Layer) to the basal surface of the Polychrome layer where the prehistoric paintings are located. This implies that there are direct interconnections that facilitate the exchange of fluids between the exterior and the interior of the cave.

Among the vertical discontinuities, the central fracture stands out. It runs close to the surface (lapiés area), from approximately 1.4 m depth to the basal surface of the Polychrome Hall, as shown in Figure 9.

Discontinuities in the Dolomitic layer, which is separated from the Polychrome layer by a thin loamy–clay intercalation [8], reduce the degree of waterproofing at the contact of these two layers, thus enhancing dripping on the basal surface (Figure 10b).

In addition, the presence of internal reflectors has been detected in the Polychrome layer, between 24 cm and 47 cm deep (23 cm thick), in the profiles measured using a 900 MHz antenna (Figure 11). These reflectors may correspond to the internal characteristic of hydroplastic deformation in the Polychrome layer.

## 4. Discussion

The water supply to the Polychrome Hall comes from direct infiltration of rainwater through the fractures. Knowing its regime is fundamental to the conservation of the cave. The availability of a detailed elevation model of the layer overlying the Polychrome Hall has made it possible to use hydrological tools to determine the size of the drainage basins, watercourses, water accumulation zones, and potential sinkholes through which water enters the underlying layer.

Having an accurate model of the interior and exterior relief of the cave is of vital importance for several reasons. First, the exact location of the strata and the geological features within the overlying layer is essential to knowing the state of karstification of the cave and the causes of the deterioration of the rock art within it. However, it is also fundamental to guarantee the safety of the workers and visitors to the cave since it lets us know and mitigate geotechnical risks such as collapse or subsidence and to establish tracking methods in the areas that require it. In addition, the models make it possible to locate the results of future, present, and even past studies, making it possible to make long-term evolution and strategic decisions on conservation. Finally, in environmental terms, these models help with the evaluation of the impacts of human activity on the conservation of the cave, making it possible to propose effective environmental and preventive conservation mitigation measures.

The results of this study have confirmed the existing interconnection between the exterior and interior environments of the Polychrome overlying layer, mainly due to the existence of vertical fractures that run from the exterior surface to the ceiling, as is the case for the most significant central fracture. These vertical fractures interconnect different discontinuities (fractures, joints, detachments, and stratigraphic planes) that lead to the circulation of gases and the infiltration of water via fracturing and/or open joints, i.e., there are interconnections between the discontinuities in the quarry through which processes of infiltration and fluid circulation and exchange occur between the cavity (Polychrome Hall) and the outside.

Using antennas with 400 MHz and 900 MHz frequencies inside the Polychrome Hall let us map the main discontinuities and develop a detailed description of the internal stratigraphy in the geological environment near the basal surface where the prehistoric paintings are located. Disruptions within the Dolomitic layer are a clear sign of alterations or disintegration, decreasing the waterproofing effect, segregated from the Polychrome layer by slim intercalation of loamy-clay, diminishing the waterproofing effect at the juncture of these two strata. This amplifies droplet formation on the lower surface.

Thus, the GPR dataset has shown that the spatial arrangement of discontinuities in the layer overlying the Polychrome Hall is complex. For further analysis of the conclusions obtained in this study, it would be convenient to conduct a 3D GPR study, reducing the distance between the profiles of the ceiling either in certain areas or in its entirety. This would have a significant impact on the cohesion of data associated with the zonation of wetting in the lithological levels, the internal structure of the central fracture, filling/opening processes of discontinuities (fractures/joints), relationship of the fracturing of the Dolomitic layer with drip points and the network of basal microchannels, and condensation processes, among others.

## 5. Conclusions

The periodic control to which areas of the ceiling of the Polychrome Hall are subjected has made it possible to detect and record direct alterations in the paintings associated with the washing and erosion by the dissolution of its mineral parts. Scanning Electron Microscopy (SEM) shows detachments of particles of millimetric order that tell us about flaking, disintegration, disaggregation, or loss of cohesion. This loss of adhesion of the pigment to the support is associated with the physical–chemical features of water, which incorporates CO_2_ and other nutrients from the soil [8] to the surface of the ceiling through the outer soil cover, partially dissolving the limestone rock [8].

In summary, one cause of the fall and loss of pigment is the partial and/or selective dissolution of the surface of the limestone support of the painting associated with the physical and chemical features of one of the most important agents of alteration [9] present in the Polychrome Hall: water.

A GPR survey lets us promote a study of the microenvironmental dynamics in relation to the processes involved in the CO_2_ fluxes between the cave and the exterior.

Continuing with this analysis and focusing on other possible causes of the loss of pigment that could cause acceleration in the dragging and washing of paint, it is worth noting the presence, not constant or permanent, of more surface moisture as small laminar flows of water concentrated in the central part of the ceiling of the Polychrome Hall on both sides of the central fracture, about 70 cm in both north and south directions.

The combination of GPR and geomatics techniques carried out in the Altamira cave, both from the top and from inside the Polychrome Hall, have shown that the central fracture has a vertical development that begins on the surface (lapiés zone) near the Polychrome Hall ceiling, giving the central fracture an average dimension of about 7.5 m in depth, constituting a direct water access route.

It will be interesting, in the future, to evaluate the state of the overlying layer from the dolomitic level to the Polychrome level using a reliable and non-destructive solution based on a higher frequency GPR antenna in the range of 1 GHz–2 GHz. The objective will be a high-resolution 3D survey to determine the trajectory followed by water from the dolomitic layer to the Polychrome layer, allowing the calculation of thicknesses and identification of the existence of possible voids, zones of presence/absence of water, cracks, and fractures.

This conservation work has a priority objective that is crucial to maintaining the cave and the art it houses in the best possible condition. The valuable information obtained through these geophysical prospecting studies combined with geomatics techniques deepens our knowledge of the causes of deterioration, which are still active today, to establish lines of research that will help us reduce the risks of deterioration and reduce or slow down its impact on the art in the cave.

The study presented here enhances our understanding of the dynamics of the water that has accessed and continues to access the Polychrome Hall’s ceiling. It, along with related research on water dynamics, enables the development of a conservation strategy focused on putting indirect measures into practice to proactively prevent damage by eliminating the risk of alteration through the avoidance of water’s solvent effect on the rock art. Detailed knowledge of water pathways allows for indirect interventions to divert specific water flows responsible for carrying away small pigment particles due to occasional water drips.

However, due to the complexity of Altamira’s conservation challenges, which are largely influenced by anthropogenic factors, there is no alternative conservation strategy but to maintain maximum internal microenvironmental stability, reducing material and energy exchange rates. Our efforts, through this study, are geared towards determining the speed of alteration processes and trying to slow them down.

## Figures and Tables

**Figure 1 sensors-23-09153-f001:**
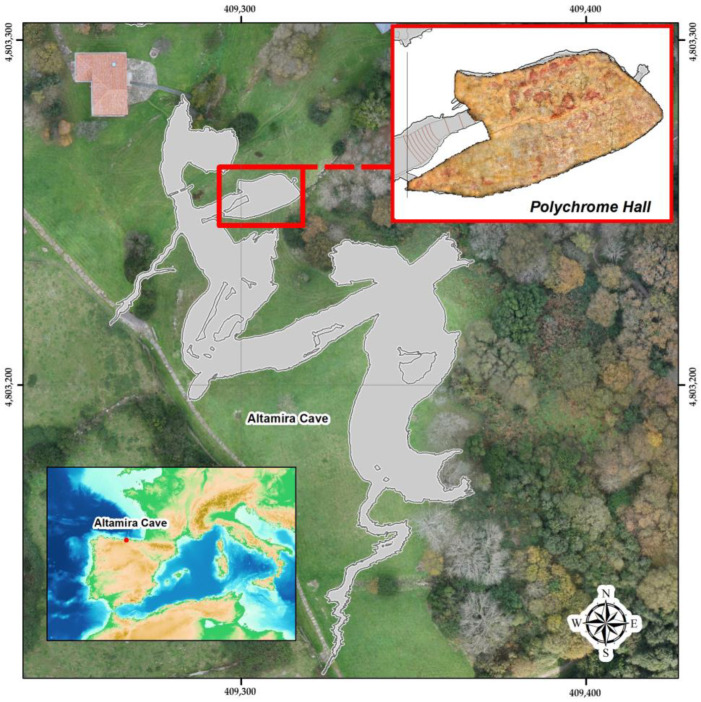
Geographic location of Altamira cave (Cantabria, Spain) and location of the main elements in the surrounding area.

**Figure 2 sensors-23-09153-f002:**
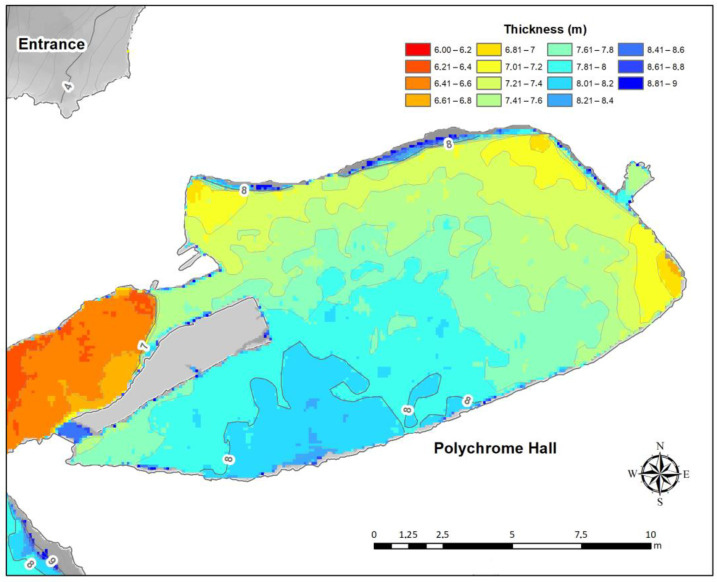
The thickness of the layer overlying the Polychrome Hall.

**Figure 3 sensors-23-09153-f003:**
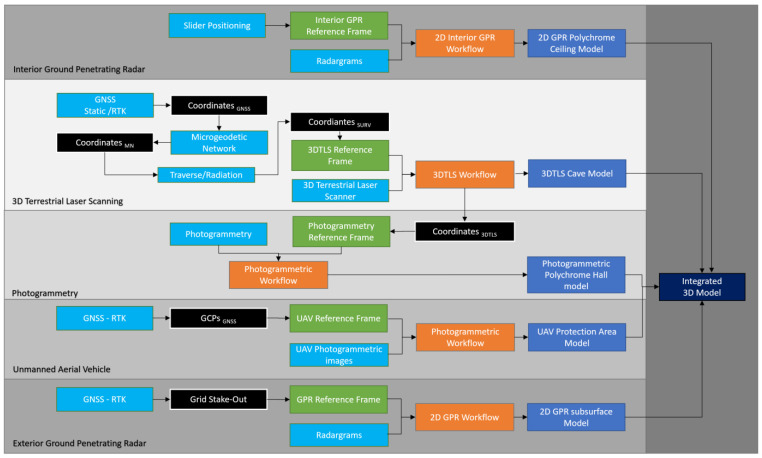
The general workflow followed in this study (adapted from [56]).

**Figure 4 sensors-23-09153-f004:**
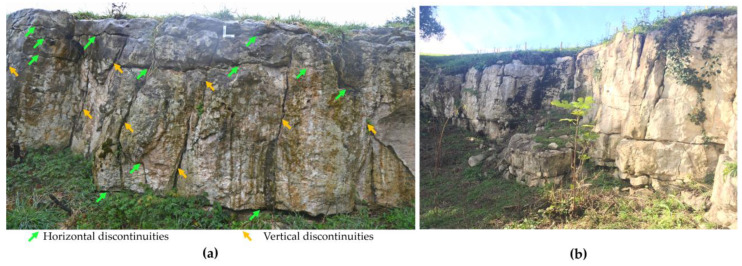
(**a**) Frontal photograph of the eastern margin of the outcrop of the layer overlying Polychrome Hall. Different vertical (yellow) and horizontal (green) discontinuities are observed in the rock mass. (**b**) Photograph showing the pattern of discontinuities throughout the outcrop of the overlying layer.

**Figure 5 sensors-23-09153-f005:**
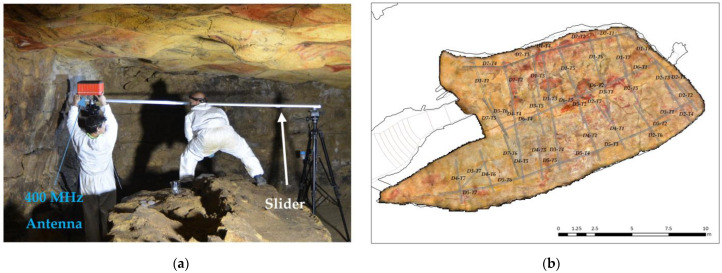
(**a**) Photograph showing the slider device used for GPR profiling. (**b**) Locations and names of the GPR profiles were measured using the 900 MHz and 400 MHz antennas projected on the Polychrome Hall ceiling where the central fracture and the Palaeolithic paintings on its surface were observed.

**Figure 6 sensors-23-09153-f006:**
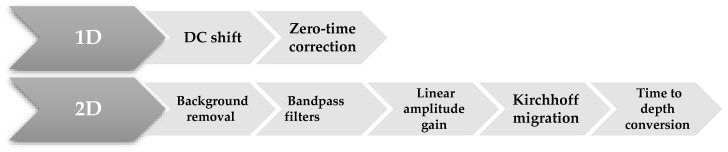
Flowchart of the main processing steps applied to raw reflection GPR data in this study.

**Figure 7 sensors-23-09153-f007:**
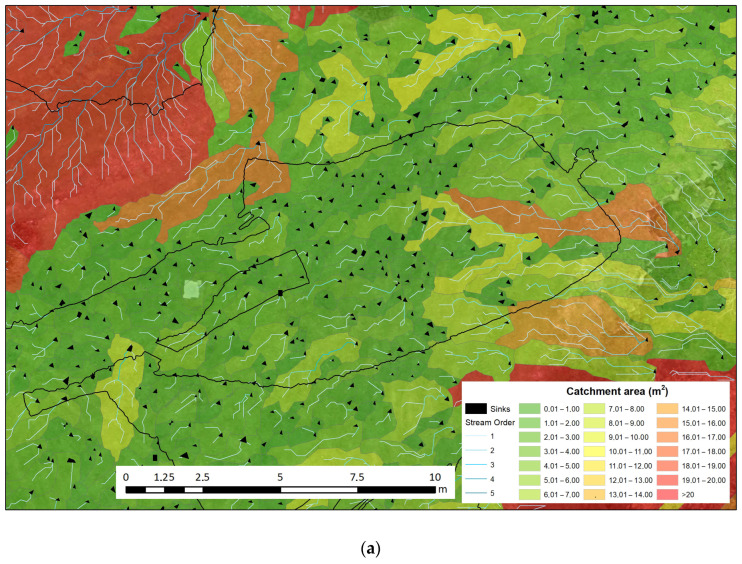
(**a**) Catchment basins classified by area and stream order according to the Strahler method [77] and vertical sinks in the Polychrome Hall. (**b**). Overlay of the above with the orthoimage and position of the recorded interior GPR profiles.

**Figure 8 sensors-23-09153-f008:**
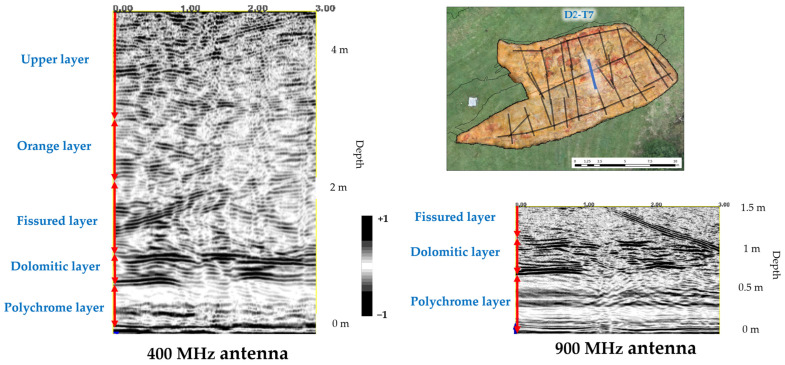
Radar stratigraphic interpretation of the layers overlying the Polychrome Hall (D2-T7 profile). The vertical red arrows indicate the thickness of each layer recorded on the radargram according to the descriptions of the overlying layer thicknesses [8].

**Figure 9 sensors-23-09153-f009:**
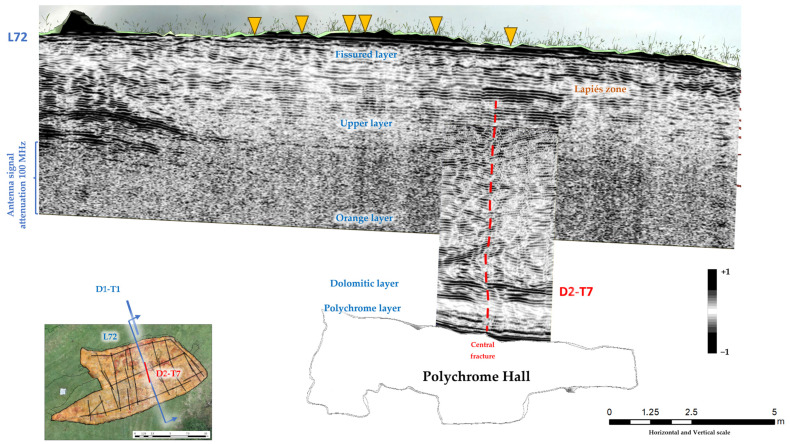
An example of mapping of the main vertical discontinuities obtained via precise georeferencing of the inner reflection profile D2-T7 (400 MHz antenna) and the outer reflection profile (100 MHz antenna). The central fracture goes from the lapiés zone (Fissured layer) to the basal surface of the Polychrome layer (indicated by the red dashed line). The surface sinks that coincide with the layout of these profiles are plotted (indicated by ochre triangles). The signal attenuation zone determined using the 100 MHz antenna and the correlations between both reflection profiles and the stratigraphic units of the layer overlying the Polychrome Hall are also shown.

**Figure 10 sensors-23-09153-f010:**
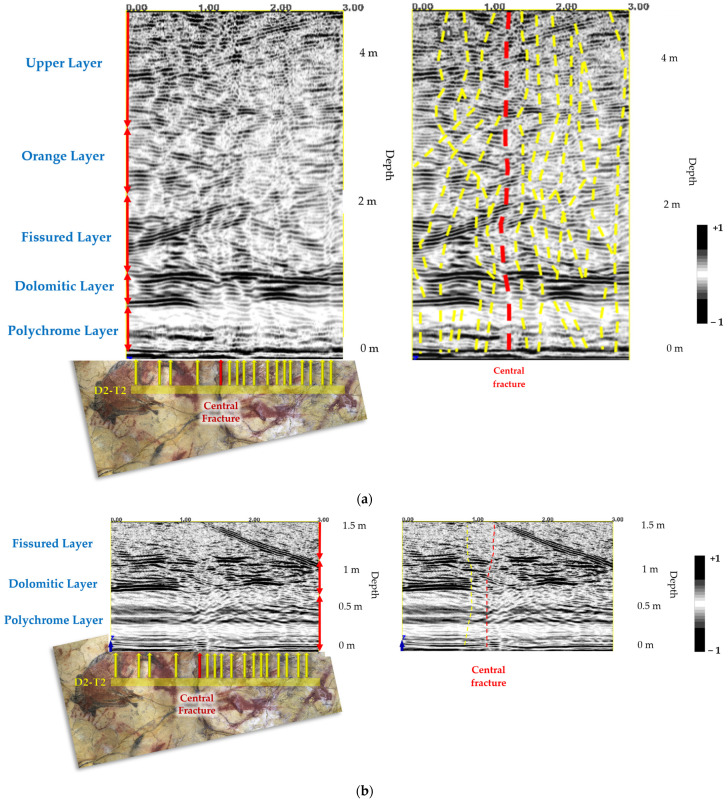
Profile D2-T7 interpretation of the main vertical discontinuities with the corresponding stratigraphy of the layer overlying the Polychrome Hall using (**a**) 400 MHz antenna and (**b**) 900 MHz antenna. In each of the two GPR records collected, the following are indicated: the position of the D2-T7 profile projected onto the orthoimage of the Polychrome Hall ceiling is shown using a yellow line; the thicknesses of each recorded layer are indicated by red vertical arrows; the location correlations between the main ceiling fractures and the records of the two antennas used are indicated by yellow arrows and the central fracture is highlighted with a red arrow.

**Figure 11 sensors-23-09153-f011:**
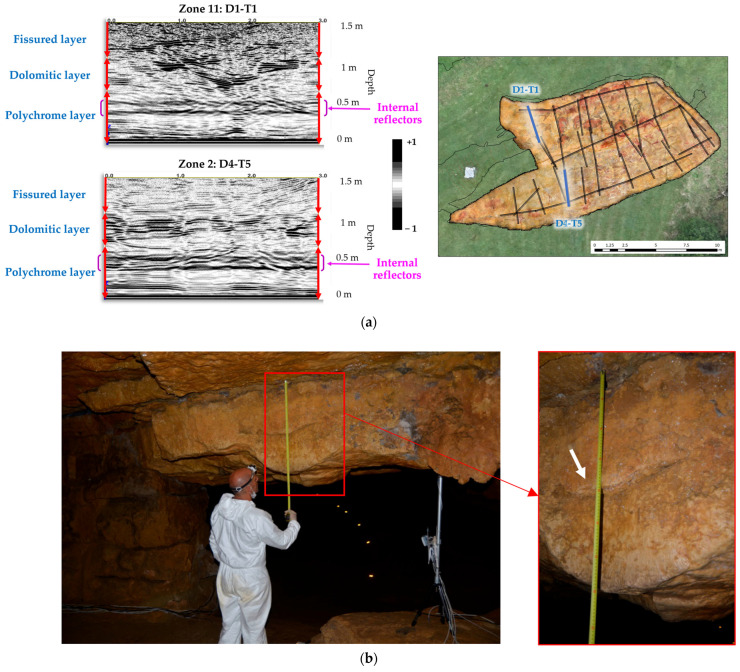
(**a**) Radar records with their stratigraphic interpretations (profiles D4-T5 and D1-T1, 900 MHz antenna) where the presence of internal reflectors in the Polychrome layer has been indicated. (**b**) Photographs of the front of the Polychrome layer at the junction before the access stairs to the Polychrome Hall showing (using a white arrow) an area of the front of the Polychrome layer that coincides with the internal reflector detected in terms of depth and thickness from the basal surface.

**Table 1 sensors-23-09153-t001:** Comparison of parameters of the 400 MHZ and 900 MHZ antennas used on the Polychrome Hall ceiling.

Parameter	400 MHz Antenna	900 MHz Antenna
Central Frequency	400 MHz	900 MHz
Frequency Range	250–600 MHz	600–1200 MHz
Wavelength in air	0.75 m	0.33 m
Wavelength (ε = 7.5)	0.306 m	0.136 m
Minimum resolution (ε = 7.5)	0.099 m	0.044 m
Estimated maximum depth (ε = 7.5) N = 20.	1.026 m	0.456 m
Estimated maximum depth (ε = 7.5) N = 100.	5.13 m	2.28 m
Ground Conditions	Less affected by challenging conditions.	Moderate attenuation under these conditions.
Advantages	Deep penetration, versatile.	Balanced depth and resolution.
Disadvantages	Lower resolution, less suitable for shallow applications.	Reduced depth compared with 400 MHz.

## Data Availability

The research data supporting this publication are not publicly available. The data were collected by GIM Geomatics as part of the research and conservation studies of the Cave. These data are kept in the Museo Nacional y Centro de Investigación de Altamira.

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
