# Peer review of "A Multimodal Research Approach to Assessing the Karst Structural Conditions of the Ceiling of a Cave with Palaeolithic Cave Art Paintings: Polychrome Hall at Altamira Cave (Spain)"

_sensors, 2023, doi:10.3390/s23229153_

Round 1

Reviewer 1 Report

Comments and Suggestions for Authors

In this paper, the authors used multiple techniques to characterize a karst cave. The organization of this paper should be improved. The results should be made more clear. Can the proposed method lead to a 3D model of the cave and the surveyed area?

1) The introduction is too long. Too much background information about the cave was introduced. Literature review of the techniques is insufficient.

2) Some background information, e.g., Line 148-149, Line 167-170, should be moved to the introduction part.

3) Figure name is mssing in Page 6.

4) In Eq.(1), epsilon should be epsilon_r, regarding to the relative permittivity.

4) In Figure 2, the method was supposed to produce a 3D model, yet the results were just an analysis of 2D GPR profiles. Is it possible to construct a 3D model of the cave?

Author Response

Dear Reviewers,

We would like to thank the editor for the time and effort to gather insightful reviews for our submission. We also would like to thank the editor and the reviewers for their valuable comments on the manuscript that help significantly improve our paper's quality in this revision.

Based on the provided set of reviews, we have carefully revised our paper. The details of our revision are described below. We hope that the reviewers find the changes satisfactory, and the revised manuscript successfully addresses the comments of the reviewers.

The revised manuscript is submitted to the MDPI submission website, and we attach this letter that discusses our changes made with respect to each of the comments. Again, these reviews were very instructive, and we would like to thank the reviewers once again for their time to review the paper.

You can find the point-by-point answers in the attached document

Best regards,

The authors

Reviewer 2 Report

Comments and Suggestions for Authors

Dear authors,

the article is very interesting and show the usefulness, but above all the problems and difficulties, of GPR prospection in complex environmental.

I have only to remark some "typos":

- Fig. 4a: the bottom yellow arrow that mark the bottom horizontal fractur have to be green, not yellow.

- p. 8, line 277:  "Several post-acquisition radar signal process techniques that can be used for GPR.:::": I think that the higlithed word "that" has to be deleted.

- p. 12, line 382, "(figure 9)": i think that you referred really to fig. 11, isn't it?

- ther isn't any reference in the text to figure 10: i think that this typo can be resolved putting the reference to the figure between the lines 408-411.

Comments on the Quality of English Language

I consider that the overall quality of the english language is good.

Reviewer 3 Report

Comments and Suggestions for Authors

1. Background about the culture/era that created the paintings should be added. Many readers will not have a background in Paleolithic.

2. A slightly more involved discussion of the local environmental and geological conditions should be added.

3. What was the effect of the air gap between the antenna and the ceiling? Did the decoupling of the antenna with surface impact signal depth penetration? Did it create more airwaves in the data from the walls, overhangs, etc...? This should be addressed.

4. Did RADAN automatically conduct a dewow or DC shift on traces? I believe it does...may want to add. 

5. Was there any preprocessing in the field? What parameters were entered into the SIR3000 for each frequency? That should be included. 

6. All maps need scale bar.

7. Figure 11 is amazing, and really shows the power of accurate exterior and interior mapping.  

8. Authors should consider a direct comparison of the data from the 400 MHz and 900 MHz. Even a summary table would be helpful.

9. Authors might consider adding an additional paragraph after the final paragraph in the Conclusion addressing the specific conservation techniques a little more explicitly. Just a suggestion. 

10. Another interesting pathway for research (not necessarily for this paper) would be aggregating the GPR data collected from each frequency inside the cave and extracting highest amplitude frequencies from across the ceiling to assess variability in absorption rates and evaluate moisture conditions in the rock. This could help prioritize conservation efforts to specific areas. 

Author Response

(The authors gave the same response as above.)

Reviewer 4 Report

Comments and Suggestions for Authors

Original Submission

Recommendation

Major revision

Comments to Author:

Title:   

A multimodal research approach for the knowledge and assessment of the karst structural conditions of the ceiling of a cave with Paleolithic cave art paintings: Polychrome Hall at Altamira cave (Spain). 

Overview and general recommendation. 

This paper is demonstrating a routine for Radar/lidar usage in earth science, and it is very good work. It is also very accurate and important; anyhow it is a very hard job to do, and the technique must be applied and controlled very precisely. These kinds of data are always welcomed to MDPI journals including RS, Sensors, Imaging etc. I really like the material: it is very advanced and (very) interesting. I like this paper very much (very good job!). The abstract and introduction is perfect. The refs are very good in introduction. The English is fair. The authors show that they know how to do the technique, and I am very glad for the routine that they put in the net.

The methodology and data employed in this paper are exemplary. I believe this research is of high quality and ready for publication, though I have reservations about its suitability for MDPI-Sensors as the journal choice.

The quality of figs are a bit low, pls improve them!

I think this paper must be go to a journal with Impact FACTOR of MAX=1.5. However the work is very good, but explanation is a bit week.

Comments on the Quality of English Language

The English is fair. 

Author Response

Dear Reviewer,

We sincerely appreciate your thoughtful and constructive feedback on our paper. Your positive assessment of our work, particularly your recognition of the importance and accuracy of our research, is truly encouraging. We are also grateful for your kind words about our methods and data, which reflect the dedication and effort we've put into this study.

Altamira Cave, stands as a paramount archaeological and artistic treasure. This site serves as an exceptional testament to prehistoric cave art, housing paintings dating back approximately 20,000 years. However, the conservation of the Altamira Cave has posed a perennial challenge. The cave is susceptible to natural erosion, humidity, and temperature variations, all of which jeopardize the stability of the paintings and rock formations. Moreover, human impact stemming from years of visitor influx has contributed to site degradation. Management of conservation and visitor access to the cave has been a subject of controversy, leading to its temporary closure to the public. The cave is closed to the public and this work has contributed to understanding the causes of such deterioration.

We greatly appreciate your feedback and your insights regarding the suitability of our paper for MDPI-Sensors. Based on your valuable input, we believe our work aligns well with the scope of the Special Issue "Advances in Ground Penetrating Radar Applications: New Developments in Acquisition and Data Analysis." And fits well their keywords ground-penetrating radar, GPR signal processing, GPR technology, and GPR data acquisition. Your insights are invaluable to us, and we have refined our work to meet the standards of the journal.

Given this, we kindly ask for you to consider our paper to include in this Special Issue. We are confident that our research will contribute to the topic, and we look forward to the opportunity to share our findings with a relevant and interested audience.

We have integrated as well the comments of the other reviews and we have tried to improve figures by adding some more information.

Once again, thank you for taking the time to review our paper and for your kind words of encouragement. Your feedback will undoubtedly help us enhance the quality of our research further. We look forward to potentially working towards a successful publication.

Sincerely,

The authors

Round 2

Reviewer 1 Report

Comments and Suggestions for Authors

All questions have been properly addressed, and the revisions are acceptable.

Author Response

Dear Reviewer,

We would like to express our gratitude for your invaluable review of our scientific paper entitled "A multimodal research approach for the knowledge and assessment of the karst structural conditions of the ceiling of a cave with Paleolithic cave art paintings: Polychrome Hall at Altamira cave (Spain). ". Your thoughtful comments and suggestions have been instrumental in improving the quality of our work. We are pleased to inform you that we have carefully considered your feedback and have made the necessary revisions to address all of your concerns.

Your positive feedback that "All questions have been properly addressed, and the revisions are acceptable" is greatly appreciated, and we are pleased to hear that our revisions have met your expectations. Your review has been a crucial step in the peer-review process, and your expertise has significantly contributed to the refinement of our paper.

We believe that your input has enhanced the clarity, accuracy, and overall impact of our research. Your detailed feedback has been a valuable asset to our work, and we are thankful for your dedication to upholding the quality of scientific publications.

We are committed to the advancement of our field and are thankful for the support and guidance from reviewers like you. We believe that your insights have strengthened the scientific community's access to our findings and have enriched the knowledge in our area of study.

Once again, we would like to express our appreciation for your time and effort in reviewing our paper. Your professional evaluation has been a significant contribution to the success of our research. If you have any further comments or questions, please do not hesitate to contact us.

Thank you for your invaluable assistance in this process.

Sincerely,

The authors

Reviewer 4 Report

Comments and Suggestions for Authors

As mentioned in my previous review, this paper demonstrates significant merit and showcases commendable efforts.

Comments on the Quality of English Language

English is OK and small corrections are needed!

Author Response

Dear Reviewer,

We would like to express our sincere appreciation for your constructive feedback and your continued support of our work. Your comments in the previous review were not only encouraging but also provided valuable insights that have significantly contributed to the improvement of our manuscript.

We are pleased to hear that you found our paper to demonstrate significant merit and commendable efforts. It is always our aim to deliver research of the highest quality, and your positive assessment is a source of motivation for us.

In light of your previous review, we have carefully considered your suggestions and addressed the necessary revisions to further enhance the clarity and quality of our paper. Your guidance has helped us refine our research and strengthen the overall impact of our findings. We believe that your feedback has played a crucial role in making our paper more robust and informative.

We are grateful for your time and expertise in reviewing our work and for your continued support in advancing scientific knowledge. Your commitment to fostering excellence in research is invaluable, and we are committed to implementing the changes you have suggested to ensure the paper meets the highest standards.

Thank you once again for your dedication to the peer review process, and we look forward to your feedback on the revised manuscript. Your input has been instrumental in shaping the final version of our paper, and we believe it will contribute positively to the scientific community.

Sincerely,

The authors